# FlowDCN: Exploring DCN-like Architectures for Fast Image Generation with Arbitrary Resolution

**Shuai Wang**
Nanjing University

**Zexian Li**
Alibaba Group

**Tianhui Song**
Nanjing University

**Xubin Li**
Alibaba Group

**Tiezheng Ge**
Alibaba Group

**Bo Zheng**
Alibaba Group

**Limin Wang** ✉
Nanjing University, Shanghai AI Lab

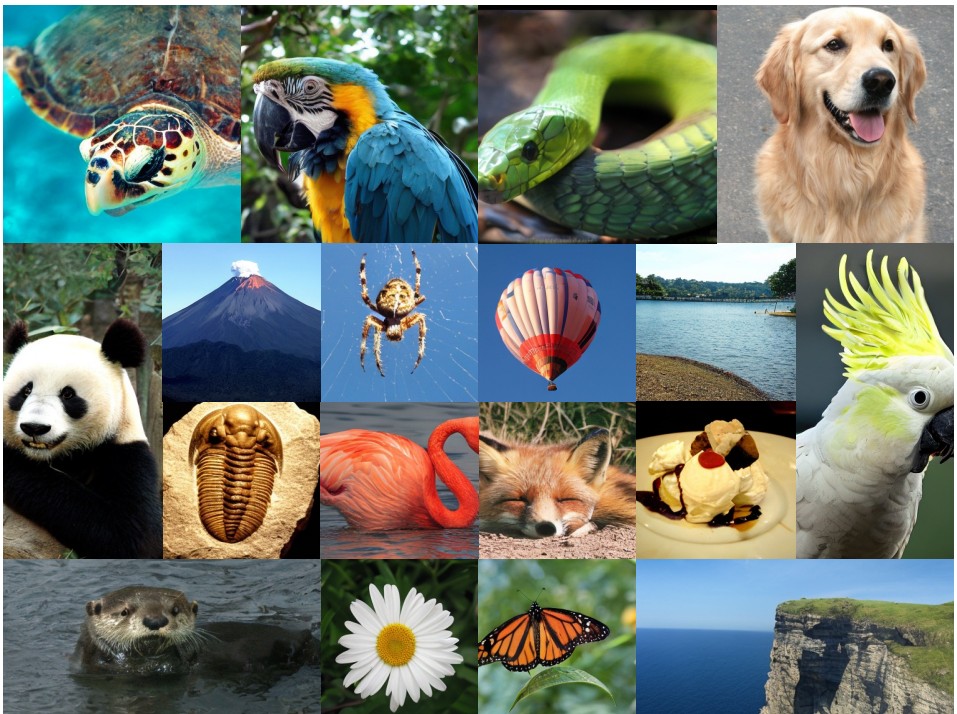

Figure 1: **Selected arbitrary-resolution samples (384x384, 224x448, 448x224, 256x256).** Generated from a single FlowDCN-XL/2 model trained on ImageNet 256×256 resolution with CFG = 4.0.

## Abstract

Arbitrary-resolution image generation still remains a challenging task in AIGC, as it requires handling varying resolutions and aspect ratios while maintaining high visual quality. Existing transformer-based diffusion methods suffer from quadratic computation cost and limited resolution extrapolation capabilities, making them less effective for this task. In this paper, we propose FlowDCN, a purely convolution-based generative model with linear time and memory complexity, that can efficiently generate high-quality images at arbitrary resolutions. Equipped with a new design of learnable group-wise deformable convolution block, our FlowDCN yields higher flexibility and capability to handle different resolutions with a single model. FlowDCN achieves the state-of-the-art 4.30 sFID on $256 \times 256$ ImageNet Benchmark and comparable resolution extrapolation results, surpassing transformer-based counterparts in terms of convergence speed (only $\frac{1}{5}$ images), visual quality, parameters (8% reduction) and FLOPs (20% reduction). We believe FlowDCN offers a promising solution to scalable and flexible image synthesis.

38th Conference on Neural Information Processing Systems (NeurIPS 2024).

# 1 Introduction

Image generation is an important task in computer vision research, which is aimed at capturing the inherent data distribution of original image datasets and generating high-quality synthetic images through sampling. Diffusion models [1, 2, 3, 4, 5] have recently emerged as a highly promising foundation for training algorithms in image generation, outperforming GAN-based models [6, 7] and Auto-Regressive models [8] by a significant margin. The evolution of diffusion models is fast, transitioning from discrete forms [1] to SDE-based continuous forms [2, 3, 4, 5, 9]. In a nutshell, diffusion models incrementally degrade an image through a time-dependent stochastic perturbation process and then learn the reverse process to restore the original image from its corrupted state.

Beyond theoretical advancements in diffusion models, the architecture of these models also significantly influences the quality of generated images. Many works [1, 10, 11] in the diffusion domain adopt a standard UNet architecture as the generation backbone, which consists of downsample blocks, upsample blocks, and long residual connections between these components. Inspired by the success of the vision transformer in perception tasks, DiT [12] eliminates the long residual connection in favor of a pure transformer-based architecture. Through rigorous experiments, DiT demonstrates that the UNet inductive bias is not essential for achieving high performance in diffusion models [12]. Meanwhile, PixArt [13, 14] and SD3 [15] venture further by significantly increasing the number of parameters, exploring new frontiers in model architecture and its impact on image generation.

When considering the generation of images at arbitrary resolution, diffusion transformers need to confront at least two primary challenges. The first is the *quadratic computation cost*: the architecture of diffusion transformers employs attention mechanisms to aggregate spatial tokens. Owing to the dense nature of attention computations, high-resolution image generation inevitably leads to significant computation and memory demands, both scaling with $O(n^2)$ complexity. To address the quadratic computation challenge, some methods [16] have adapted recurrent computational strategies from natural language processing. However, these adaptations do not fully capitalize on the strengths of autoregressive tasks and result in slower inference speeds due to the reduced parallelism inherent in RNN-based scanning. The second challenge is *resolution extrapolation*: many diffusion transformers rely on absolute position embedding (APE) [17] to incorporate positional information, introducing it at the onset of the model. This approach forces subsequent layers to become overfitted to the APE for providing positional context to the attention layers, which presents a significant barrier when extrapolating to different resolutions. To address this issue, FiT [18] has turned to Rotary Positional Encoding [19], incorporating RoPE2D to enhance its resolution extrapolation capabilities. Nevertheless, FiT still requires a training pipeline tailored to arbitrary-resolution generation.

In contrast, convolutional models are the most common choice of visual encoders, boasting linear complexity and aggregating spatial features based on relative positions. With the support of modern convolution operators [20, 21, 22], convolutional models have demonstrated comparable performance or even surpassed transformers in perception tasks. This naturally leads us to inquire: *Can modern convolutional networks achieve arbitrary-resolution generation efficiently and outperform transformer counterparts?* To answer this question, we opt for deformable convolution as the basic block for exploration in generation, owing to its superior performance in perception tasks.

Specifically, we propose a novel approach to decouple the scale and direction prediction of deformable convolution, giving rise to a group-wise multiscale deformable convolution block that enables efficient multiscale feature aggregation. By leveraging this block, we introduce FlowDCN, a modern purely convolution generative model that tackles arbitrary-resolution generation. Thanks to the new design of convolutional deformable block, our FlowDCN yields higher flexibility and capability to handle different resolutions with a single model. The experiments demonstrate that FlowDCN consistently surpasses its diffusion transformer counterparts, DiT [12] and SiT [23]. Notably, on the 256x256 ImageNet benchmark, FlowDCN achieves faster convergence, yielding SoTA sFid of 4.30 and FID of 2.13 under 1.5M steps with batch size 256, while exhibiting 20% lower latency, 8% fewer parameters, and approximately 20% fewer floating-point operations (FLOPs). On the 512x512 ImageNet benchmark, FlowDCN achieves 4.53 sFid o and 2.44 FID under 100K finetuning steps with batch size 256.

Moreover, our FlowDCN offers a significant advantage in fast arbitrary-resolution generation, as it only requires linear time and memory complexity. Through visualization comparisons, our FlowDCN demonstrates substantially better visual quality even at extremely small sampling steps, such as 3, 4, and 5 steps. To further enhance its visual quality, we propose Scale Adjustment, a simpler

technique for extrapolating resolution to unseen dimensions. Our results show that FlowDCN achieves comparable resolution extrapolation capabilities to highly tailored methods, underscoring its potential for generating high-quality images at various resolutions. The contributions can be summarized as:

- We decouple the scale and direction priors of deformable convolution and propose a Group-wise MultiScale Deformable Block. Building upon this block, we propose FlowDCN, a purely convolution-based generative model with high efficiency.
- On 256x256 ImageNet benchmark, under only 1.5M training steps, our FlowDCN-XL/2 achieves 2.13 FID and SoTA 4.30 sFID with Euler solver and classifier free guidance.
- On 512x512 ImageNet benchmark, under only 100K finetuning steps, our FlowDCN-XL/2 achieves 2.44 FID and 4.53 sFID with Euler solver and classifier free guidance.
- We propose a much simple and efficient resolution extrapolation method, deemed as Scale Adjustment. For arbitrary resolution generation, we achieve comparable results to highly tailored methods.

## 2 Preliminary

### 2.1 Linear-based Flow Matching

Flow matching [4, 5] is a simple but powerful diffusion family. We incorporate linear-based flow matching as the training framework for its simplicity. Given the image sampled $x$ from training distributions and the noise $\epsilon$ sampled from a Gaussian distribution, linear-based flow matching forward process interpolate $x_t$ with $x$ and $\epsilon$ using the following equation:

$$x_t = tx + (1 - t)\epsilon. \tag{1}$$

The velocity field of linear-based flow matching [4, 5] is defined as Eq. (2). We train our FlowDCN to predict the time-dependent velocity field between $x$ and $\epsilon$:

$$v_t(x_t) = x - \epsilon. \tag{2}$$

During training, the flow matching objective directly regresses the target velocity:

$$\mathcal{L}_v = \int_0^1 \mathbb{E}[\| v_\theta(x_t, t) - v_t(x_t) \|^2]dt. \tag{3}$$

For sampling, the common ODE/SDE solver *e.g.*. Euler method, Heun method can be employed.

### 2.2 Deformable Convolution Revisited

Given an image feature $\mathbf{x} \in \mathbb{R}^{H \times W \times D}$, deformable convolution predicts the deformable field $\Delta\mathbf{P}(\mathbf{x}) \in \mathbb{R}^{H \times W \times G \times K \times 2}$ and the dynamic weights $\mathbf{W}(\mathbf{x}) \in \mathbb{R}^{H \times W \times G \times K}$ from the image feature $\mathbf{x}$. Specifically, $H$ and $W$ represent the height and width of the feature spatial shape, $D$ is the feature channel, $K$ is the number of sampling points, and $G$ is the number of groups in the deformable convolution operation. The deformable field and dynamic weights are computed as Eq. (4):

$$\Delta\mathbf{P}(\mathbf{x}) = \mathbf{W}_{\text{deformable}}^T\mathbf{x} + \mathbf{b}_{\text{deformable}}, \tag{4}$$

$$\mathbf{W}(\mathbf{x}) = \mathbf{W}_{\text{weight}}^T\mathbf{x} + \mathbf{b}_{\text{weight}}. \tag{5}$$

For a specific group $g$ in deformable convolution, the sampling position is determined by the base feature position $p_0$, sampling position prior $p_k$, and predicted deformable $\Delta p_k$ from $\Delta\mathbf{P}(\mathbf{x})$ for the $k$-th sampling point. The dynamic weight $w_k$ is provided from $\mathbf{W}(\mathbf{x})$. The deformable convolution aggregates $K$ sparse spatial features according to the sampling location and dynamic weight as following:

$$\mathbf{y}^g(p_0) = \sum_{k=0}^{K} w_k^g \mathbf{x}^g(p_0 + p_k + \Delta p_k(\mathbf{x})), \tag{6}$$

$$\mathbf{y} = \text{concat}(\mathbf{y}^1, \mathbf{y}^2, ...., \mathbf{y}^G). \tag{7}$$

The predefined spatial position prior $p_k$ is initialized from the regular convolution, commonly using $(-1, -1), (-1, 0), ...(0, 0), ...(1, 1)$ as the predefined value.

Deformable convolution introduces long-range dependencies and dynamic aggregation into regular convolutions, bridging the gap between convolution and multi-head self-attention [24]. Thus, deformable convolution shares the efficiency merit of convolution and the dynamics merit of the attention mechanism. In most scenarios, DCN-like architectures are more powerful than common CNNs, we provide comparison experiments of DCN and CNN of flow matching training. Notably, deformable convolution directly predicts the dynamic weights and only aggregates limited features from spatial locations, enjoying a relatively sparse computation diagram. A deformable convolution operator only requires $\frac{4KHWC}{G}$ FLOPs for computation when employing bilinear sampling to aggregate features.

## 3 Method

### 3.1 Multi-Scale Deformable Convolution

The original deformable convolution has been widely adopted in hierarchical model architectures [25, 24, 20] for perception tasks. However, these models typically progressively downsample the feature maps to increase the reception field growth rate. In contrast, image generation tasks require outputs with more high-frequency details and low-level information. From this perspective, progressively downsampling features would lead to the loss of high-frequency details. One possible solution is to introduce long residual connections to generation models [11, 10]. However, in practice, this approach demands caching image features from the encoder part, which increases peak memory usage during model inference.

To strike a balance between receptive fields and high-frequency details, we propose decoupling the deformable field into scale and direction, and introduce a novel multiscale deformable convolution. Unlike previous deformable convolutions, our approach assigns different scale priors to different groups.

**Decoupling deformable field to direction and scale.** The original deformable convolution directly regresses the deformable field to learn an unbounded and adaptive sampling point generator. However, the vast image spatial range poses a challenge to the learning process, as it leads to unstable regression of the deformable range when extrapolating from local neighbors to distant feature locations. We tackle this problem by decoupling the direction and scale of the deformable field. Specifically, we reorganize the sampling point formulation in Eq. (9).

$$s(\mathbf{x}) = S_{\max} * \text{sigmoid}(\mathbf{W}_s^T \mathbf{x}), \tag{8}$$

$$p = p_0 + s(\mathbf{x}) * (p_k + \Delta p_k(\mathbf{x})), \tag{9}$$

where $s(\mathbf{x})$ is the learnable scale predicted from the image feature $\mathbf{x}$. $S_{\max}$ is the max scale value of the given deformable convolution, we leave it as a hyper-parameter only related to input resolution, thus we can manually tune it according to input resolution, details are placed in Sec. 3.3.

**Group-wise multi-scale deformable convolution.** To keep high-resolution feature maps and own a large reception field growth rate, we propose to assign different scale priors to different groups. This allows deformable groups with large scale priors to aggregate long-dependency features, while those with small scale priors aggregate short-dependency features as follows:

$$s^g(\mathbf{x}) = S_{\max} * \text{sigmoid}(\mathbf{W}_s^T \mathbf{x} + s_0^g), \tag{10}$$

$$p = p_0 + s^g(\mathbf{x}) * (p_k + \Delta p_k(\mathbf{x})). \tag{11}$$

Specifically, we initialize the scale priors with Eq. (12) and initialize $\mathbf{W}_s$ with zeros to obtain linearly increased $\text{sigmoid}(s_0^g)$ along group axis:

$$s_0^{g+1} = \log(\frac{g}{G-g}). \tag{12}$$

### 3.2 Flow-based Deformable Convolutional Generative Model

We introduce our novel diffusion generation architecture, dubbed FlowDCN. Rather than directly adopting tailored architectures for image generation, such as long residuals and normalization techniques, we aim to explore the generative capabilities of deformable convolution-based architectures

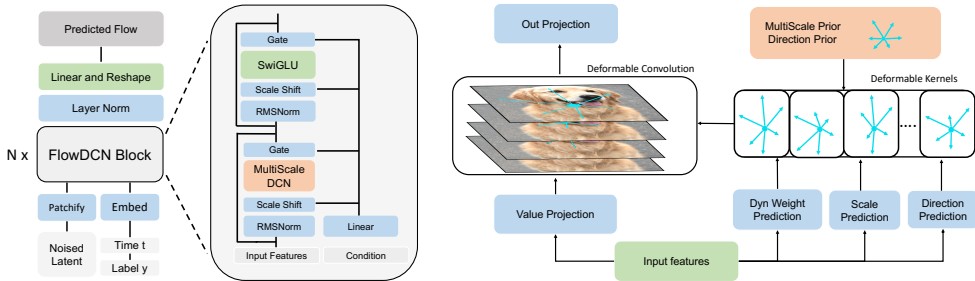

(a) **FlowDCN Architecture.** Our FlowDCN consists of stacked MultiScaleDCN blocks and SwiGLU blocks. We also employ RMSNorm to stabilize training.

(b) **MultiScale DCN Block.** Dynamic weight and scale& direction deformable field are predicted from input features, then merged with priors to form the deformable kernels to extract features.

Figure 2: **The Architecture of Our FlowDCN and MultiScale DCN Block.**

in a faithful manner. To this end, we deliberately discard long residual connections and opt to build a pure convolution-based generative model, preserving the unique characteristics of DCN-like models as much as possible. For training and sampling, we leverage the powerful flow matching algorithm to align our model with the state-of-the-art SiT [23].

**Deformable convolution generative model.** The model architecture is illustrated in Fig. 2a. We aim to build a pure DCN-like generative model to explore the generation ability of DCN-like [20] architectures. To match the base resolution of model input with DiT [12] and SiT [23], we similarly patchify the noisy input via convolution. Inspired by DiT [12], we inject the timestep and label conditions through adaLN-Zero [12, 26]. The basic block is formulated as Eq. (13). Drawing inspiration from LLaMA [27, 28], we replace vanilla FFN and LayerNorm with SwiGLU and RMSNorm, respectively. Note we also provide FFN and LayerNorm version FlowDCN for fair comparisons:

$$\mathbf{x}_1 = \mathbf{x} + \text{AdaLN}(\mathbf{y}, \mathbf{t}, \text{MultiScale-DCN}(\mathbf{x})), \tag{13}$$
$$\mathbf{x}_2 = \mathbf{x}_1 + \text{AdaLN}(\mathbf{y}, \mathbf{t}, \text{ SwiGLU}(\mathbf{x}_1)). \tag{14}$$

### 3.3 Arbitrary Resolution Sampling

We denote the training resolution as $H_{\text{train}} \times W_{\text{train}}$ and the inference resolution as $H_{\text{test}} \times W_{\text{test}}$. Notably, our FlowDCN is capable of handling arbitrary resolution that differs from the training resolution. As a reminder, the multiscale deformable convolution block aggregates features based on predicted scales and directions according to Equation (Eq. (9)). In practice, the predicted scale of the multiscale deformable convolution layer is typically fitted to match the training resolution distribution. However, this limits the reception fields of the image features when encountering unseen resolution, ultimately hurting the global semantic consistency [29, 30]. To improve the global semantic consistency, we propose adjusting the scaling factor based on the relative ratio between the training resolution and inference resolution.

**Adjust $S_{\text{max}}$ to match inference resolution.** As shown in Eq. (10), $S_{\text{max}}$ controls the maximum sampling range in multiscale deformable convolution. As discussed in Sec. 3.1, we treat it as a resolution-dependent hyperparameter. It is straightforward to observe that scaling $S_{\text{max}}$ with the relative aspect ratio between train size and inference size could match the reception field between train and inference:

$$s_h^g(\mathbf{x}) = \text{sigmoid}(\mathbf{W}_s^T \mathbf{x} + s_0^g) \cdot S_{\text{max}} \cdot \frac{H_{\text{test}}}{H_{\text{train}}}, \tag{15}$$

$$s_w^g(\mathbf{x}) = \text{sigmoid}(\mathbf{W}_s^T \mathbf{x} + s_0^g) \cdot S_{\text{max}} \cdot \frac{W_{\text{test}}}{W_{\text{train}}}. \tag{16}$$

| Operator | Runtime (ms) of Input Shape $H \times W \times G \times D$ | | | |
|---|---|---|---|---|
| | $16 \times 16 \times 16 \times 64$ | $16 \times 16 \times 16 \times 128$ | $32 \times 32 \times 16 \times 64$ | $32 \times 32 \times 16 \times 128$ |
| Attention (Math SDP) | 0.92/2.1 | 1.16/2.71 | 10.7/28.8 | 12.4/35.8 |
| Attention (Flash SDP) [34] | 0.62/N | 1.47/N | 4.98/N | 14.4/N |
| DeformConv(DCNv4 [20]) | 0.77/1.00 | 1.0/2.1 | 2.8/4.4 | 3.9/8.3 |
| DeformConv(Shm) | 0.56/0.81 | 1.1/1.5 | 2.7/3.9 | 5.0/7.3 |
| DeformConv(Triton-lang)† | 0.83/0.89 | 0.95/1.1 | 3.4/3.8 | 4.0/4.8 |

Table 1: **Op-level benchmark on standard input shape of Diffusion backbone task.** FP16/FP32 results are collected on Nvidia A10 GPU. We use 32 batch sizes for benchmarking. † indicates our Triton-lang [32] implementation of DCNv4. N indicates implementation is not available.

| Models | FID↓ | sFID↓ | IS↑ |
|---|---|---|---|
| SiT-S/2 | 7.42 | 4.47 | 8.7 |
| **FlowDCN-S/2** | **5.47** | **4.35** | **8.89** |
| w/o MultScale | 5.72 | 4.42 | 8.85 |
| w/o PriorInit | 5.68 | 4.49 | 8.9 |

| Kernel | FID↓ | sFID↓ | IS↑ |
|---|---|---|---|
| 4 | 5.88 | 4.6 | 8.89 |
| **9** | **5.47** | **4.35** | **8.89** |
| 16 | 5.39 | 4.54 | 8.93 |
| 32 | 5.13 | 4.43 | 9.05 |

| $p_k$ | $s(\mathbf{x})$ | FID↓ | sFID↓ | IS↑ |
|---|---|---|---|---|
| fixed | fixed | 5.6 | 4.58 | 8.90 |
| **fixed** | **learn** | **5.47** | **4.35** | **8.89** |
| learn | fixed | 6.01 | 4.43 | 8.85 |
| learn | learn | 5.63 | 4.37 | 8.89 |

(a) **Comparsions with SiT.** Our FlowDCN outperforms SiT by a significant margin.

(b) *KernelSize $K$ of FlowDCN.* large kernel size produces better results than small one.

(c) *Deformable fields learning setting.* Default achieves best results.

Table 2: **Ablation Studies and Comprasion with other flow-based method on 32x32 CIFAR Dataset.** In order to fully align with SiT [23], here we replace our SwiGLU and RMSNorm with FFN and LayerNorm armed in SiT, respectively. **Bold** font indicates the default setting.

# 4 Experiments

We conduct experiments on 32x32 CIFAR10 and 256x256 ImageNet datasets. The training batch size is set to 256. Similar to SiT [23] and DiT [12], we use Adam optimizer [31] with a constant learning rate 0.0001 during the whole training. We do not adopt any gradient clip techniques for fair comparison. For 32x32 CIFAR10 dataset, we train our model for 25000 steps. As for 256x256 ImageNet dataset, we train for 1.5M steps. We use $8 \times$A100 GPUs as the default training hardware.

**Efficient deformable convolution implementation.** Although DCNv4 [20] proposes a much faster deformable convolution implementation, it is not tailored for image generation input shape. For resolution below $512 \times 512$, there are fewer spatial tokens (only $16 \times 16$ tokens for $256 \times 256$ resolution) to fully utilize sparse computation strengths, thus DCNv4 exhibits even worse latency compared to attention. To remedy high latency of deformable convolution for low-resolution scenery, we decide to leverage shared memory to reduce the latency of random sampling in deformable convolutions, deemed as DeformConv(shm). We place the performance benchmark at Tab. 1. For high-resolution scenery, We also re-implement DeformConv(DCNv4) in Triton-lang as DeformConv(Triton-lang) to leverage the strengths of compiler [32, 33] to find suitable hyperparameters.

## 4.1 32x32 CIFAR Dataset

The CIFAR10 dataset[35], comprising 50,000 32x32 small-resolution images from 10 distinct class categories, is considered an ideal benchmark to validate the design of our MultiScale deformable block due to its relatively small scale. We select SiT-S/2 as a comparison baseline, as it also leverages the flow-matching framework. For sampling, we employ the Euler stochastic solver with 1000 sampling steps to generate images. We report the FID [36], sFID [37], and Inception Score [38] as the primary metrics to evaluate the performance of our model.

**Compare with baseline SiT.** We summarise the metrics of our FlowDCN and SiT in Tab. 2a. Our FlowDCN achieves 5.47 fid, surpassing its counterpart SiT with 2.0 fid margins. Additionally, our model performs slightly better in terms of sFID and Inception Scores, further demonstrating its superiority.

**Group-wise multiscale design.** As showed in Tab. 2a, we denote the variant of FlowDCN that uses vanilla deformable convolution instead of Multiscale deformable convolution as *w/o MultiScale*. Notably, the absence of group-wise multiscale deformable convolution leads to a 0.25 FID performance degradation. This result demonstrates the effectiveness and power of our proposed group-wise multiscale mechanism.

**Prior initialization.** By default, we manually initialize the direction priors with predefined grids $\{(-1, -1), (-1, 0), ...(0, 0), ...(1, 1)\}$, and initialize the scale priors with linearly increased scale along group axis. We also experiment with randomly initialized direction and scale priors in Tab. 2a, donated as *w/o PriorInit*. Random initialization shows slight performance degradation.

**Sampling points.** In Tab. 2b, We train FlowDCN with varying kernel sizes $K$ and observe that the performance consistently improves as the number increases. Specifically, using 32 points to aggregate features, FlowDCN achieves a FID score of 5.13 and an sFID score of 4.43. However, to maintain a relatively sparse pattern, we choose $K = 9$ as the default setting, striking a balance between performance and computational efficiency.

**Fixed direction priors.** In Tab. 2c, we present the results of training FlowDCN with different prior learning settings. Notably, we find that the fixed direction prior $p_k$ in Eq. (9) achieves better results compared to the learnable direction prior. We hypothesize that the learnable direction prior may cause the learning of the deformable field to become unstable, leading to inferior performance.

**Learnable relative scale.** In the $s(\mathbf{x})$ column of Tab. 2c, the notation "*learn*" indicates that we predict a relative scale of the deformable fields in addition to the learnable scale priors $s_0^g$ ($\mathbf{W}_s^T \mathbf{x}$ in Eq. (10)), whereas "*fixed*" does not predict the relative scales $s(\mathbf{x})$ in the deformable field. Learning a relative scale for each feature in Tab. 2c achieves better results of 5.47 FID.

## 4.2  256×256 ImageNet Dataset

Based on our analysis, we select the MultiScale deformable convolution with a kernel size of $K = 9$ as the basic block for our Imagenet experiments. Our default setting involves fixing the direction priors and learning relative scales from the deformable field. We manually initialize the direction priors with predefined grids and initialize the scale priors with linearly increased scales. To generate images, we employ an Euler-Maruyama solver with 250 steps for stochastic sampling. We report the FID, sFID, Inception Score, and Precision & Recall as the primary metrics to evaluate the performance of our model.

| Model | FLOPs (G) | Params (M) | Latency(ms) | FID↓ | sFID↓ | IS↑ |
|---|---|---|---|---|---|---|
| SiT-S/2 | 6.06 | 33 | 0.026 | 57.64 | 9.05 | 24.78 |
| SiT-S/2 $^\dagger$ | 6.06 | 33 | 0.026 | 57.9 | **8.72** | 24.64 |
| **FlowDCN-S/2** | 4.36 (-28%) | 30.3 (-8.1%) | 0.027 | **54.6** | 8.8 | **26.4** |
| SiT-B/2 | 23.01 | 130 | 0.084 | 33.5 | 6.46 | 43.71 |
| SiT-B/2 $^\dagger$ | 23.01 | 130 | 0.084 | 37.3 | 6.55 | 40.6 |
| **FlowDCN-B/2** | 17.87 (-22%) | 120 (-7.6%) | 0.076 | **28.5** | **6.09** | **51** |
| w/o RMS & SwiGLU | 17.88 (-22%) | 120 (-7.6%) | 0.072 | 29.1 | 6.13 | 50.4 |
| DiT-L/2 | 80.71 | 458 | 0.291 | 23.3 | - | - |
| SiT-L/2 | 80.71 | 458 | 0.291 | 18.8 | 5.29 | 72.02 |
| **FlowDCN-L/2** | 63.51 (-21%) | 421 (-8.0%) | 0.254 | **13.8** | **4.69** | **85** |
| DiT-XL/2 | 118.64 | 675 | 0.387 | 19.5 | - | - |
| SiT-XL/2 | 118.64 | 675 | 0.387 | 17.2 | 5.07 | 76.52 |
| **FlowDCN-XL/2** | 93.24 (-21%) | 618 (-8.4%) | 0.303 | **11.3** | **4.85** | **97** |

Table 3: **Image generation metrics comparisons between SiT [23], DiT [12] under 400k training steps budgets**. All metrics are calculated from the sampled 50k images under 250 Euler SDE sampling steps without classifier-free guidance. †: reproduced result. Latency(ms) is the 1-NFE latency and collected from Nvidia A10 GPU with 16 batchsize under float32.

**Metrics comparison with baseline SiT.** We present the performances of different-size models at 400K training steps in Tab. 8. From Small to XL-size models, our FlowDCN model family consistently outperforms its counterpart DiT [12] and SiT [23] with significant margins. Without RMS/SwiGLU, our FlowDCN-B/2 degrades with 0.6 FID gains but still surpasses SiT by a large margin. In addition to its superior performance and convergence speed, our FlowDCN also boasts a remarkable 8% reduction in parameters and at least 20% reduction in FLOPs compared to DiT/SiT. This demonstrates that our FlowDCN surpasses vision transformer-based generation models in multiple aspects.

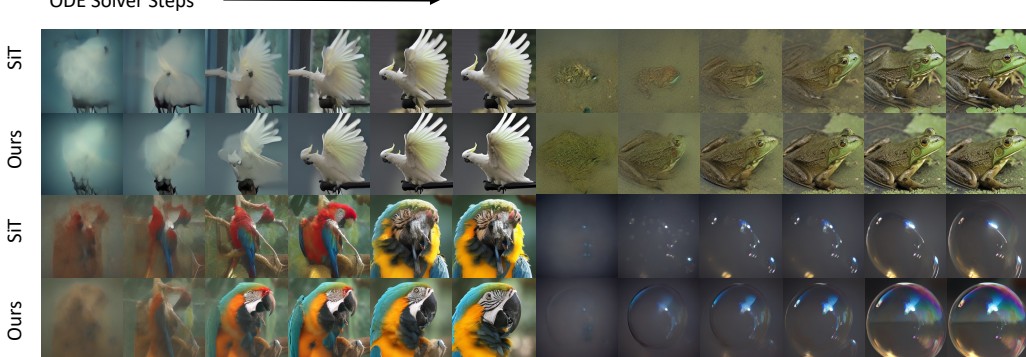

ODE Solver Steps →

SiT

Ours

SiT

Ours

Figure 3: **Visualization Comparison with SiT**. *Best viewed zoomed-in*. We sample both our FlowDCN-XL/2 and SiT-XL/2 with Euler ODE solver under 2, 3, 4, 5, 8, 10 steps using the same latent noise. At the fewer steps sampling scenery, our FlowDCN generates slightly clearer and higher-quality images.

| ImageNet 256×256 Benchmark | | | | | | | | |
|---|---|---|---|---|---|---|---|---|
| Generative Models | Long Residuals | Total Images(M) | Total GFLOPs | FID ↓ | sFID ↓ | IS ↑ | P ↑ | R ↑ |
| ADM-U [10] | ✓ | 507 | $3.76 \times 10^{11}$ | 7.49 | 5.13 | 127.49 | 0.72 | 0.63 |
| CDM [39] | ✓ | - | - | 4.88 | - | 158.71 | - | - |
| LDM-4 [40] | ✓ | 213 | $2.22 \times 10^{10}$ | 10.56 | - | 103.49 | 0.71 | 0.62 |
| DiT-XL/2 [12] | ✗ | 1792 | $2.13 \times 10^{11}$ | 9.62 | 6.85 | 121.50 | 0.67 | **0.67** |
| DiffusionSSM-XL[16] | ✗ | 660 | $1.85 \times 10^{11}$ | 9.07 | 5.52 | 118.32 | 0.69 | 0.64 |
| SiT-XL/2[23] | ✗ | 1792 | $2.13 \times 10^{11}$ | 8.61 | 6.32 | **131.65** | 0.68 | **0.67** |
| **FlowDCN-XL/2** | ✗ | **384** | $\mathbf{3.57 \times 10^{10}}$ | **8.36** | **5.39** | 122.5 | **0.69** | 0.65 |
| Classifier-free Guidance | | | | | | | | |
| ADM-U[10] | ✓ | 507 | $3.76 \times 10^{12}$ | 3.60 | - | 247.67 | 0.87 | 0.48 |
| LDM-4 [40] | ✓ | 213 | $2.22 \times 10^{10}$ | 3.95 | - | 178.22 | 0.81 | 0.55 |
| U-ViT-H/2 [11] | ✓ | 512 | $6.81 \times 10^{10}$ | 2.29 | - | 247.67 | 0.87 | 0.48 |
| DiT-XL/2 [12] | ✗ | 1792 | $2.13 \times 10^{11}$ | 2.27 | 4.60 | **278.24** | 0.83 | 0.57 |
| DiffusionSSM-XL [16] | ✗ | 660 | $1.85 \times 10^{11}$ | 2.28 | 4.49 | 259.13 | **0.86** | 0.56 |
| SiT-XL/2[23] | ✗ | 1792 | $2.13 \times 10^{11}$ | 2.06 | 4.50 | 270.27 | 0.82 | **0.59** |
| FiT-XL/2[18] | ✗ | 450 | - | 4.27 | 9.99 | 249.72 | 0.84 | 0.51 |
| **FlowDCN-XL/2 (cfg=1.375; ODE)** | ✗ | **384** | $\mathbf{3.57 \times 10^{10}}$ | 2.13 | **4.30** | 243.46 | 0.81 | 0.57 |
| **FlowDCN-XL/2 (cfg=1.375; SDE)** | ✗ | **384** | $\mathbf{3.57 \times 10^{10}}$ | 2.08 | 4.38 | 257.53 | 0.82 | 0.57 |
| **FlowDCN-XL/2 (cfg=1.375; ODE)** | ✗ | **486** | $\mathbf{4.52 \times 10^{10}}$ | **2.01** | 4.33 | 254.36 | 0.81 | 0.58 |
| **FlowDCN-XL/2 (cfg=1.375; SDE)** | ✗ | **486** | $\mathbf{4.52 \times 10^{10}}$ | **2.00** | 4.37 | 263.16 | 0.82 | 0.58 |

Table 4: **Image generation quality evaluation of and existing approaches on ImageNet 256× 256**. Total images by training steps × batch size as reported, and total GFLOPs by Total Images × GFLOPs/Image. P refers to Precision and R refers to Recall.

**Comparison with other generative models.** We report the final metrics of FlowDCN-XL/2 at Tab. 4. Our FlowDCN achieves much faster convergence speed with nearly $\frac{1}{5}$ total images compared its *No-Long-residuals* counterparts. Additionally, using Euler ODE solver and classifier-free guidance with 1.375, our FlowDCN obtains SoTA 4.30 sFID and 2.13 FID results. Training for extra 400k steps, FlowDCN will be further improved to 2.00 FID. As sFID reflects the spatial structure quality [37], better sFID shows our FlowDCN captures better structure distributions. We notice that the IS metric is lower than other models, however, there is an improvement trend along with training iterations.

**Visual quality comparison with baseline SiT.** We sample both our FlowDCN-XL/2 and SiT-XL/2 with Euler ODE solver for 2, 3, 4, 5, 8, 10 steps, employing the same latent noise for both models. Notably, at the fewer steps sampling scenario, our FlowDCN generates slightly clearer and higher-quality images. We place the generated images at Fig. 3 and Appendix.

## 4.3  512 × 512 ImageNet Dataset

As training on high-resolution images consumes much more resources, we opt to fine-tune 100k steps from the same model trained on 256 × 256 resolution setting of 1.5M steps (corresponding to

| Class-Conditional ImageNet 512×512 | | | | | |
| --- | --- | --- | --- | --- | --- |
| Model | FID↓ | sFID↓ | IS↑ | Precision↑ | Recall↑ |
| BigGAN-deep [6] | 8.43 | 8.13 | 177.90 | 0.88 | 0.29 |
| StyleGAN-XL [7] | 2.41 | 4.06 | 267.75 | 0.77 | 0.52 |
| ADM [10] | 23.24 | 10.19 | 58.06 | 0.73 | 0.60 |
| ADM-U [10] | 9.96 | 5.62 | 121.78 | 0.75 | 0.64 |
| ADM-G [10] | 7.72 | 6.57 | 172.71 | 0.87 | 0.42 |
| ADM-G, ADM-U | 3.85 | 5.86 | 221.72 | 0.84 | 0.53 |
| DiT-XL/2 [12] | 12.03 | 7.12 | 105.25 | 0.75 | 0.64 |
| DiT-XL/2-G [12] (cfg=1.50) | 3.04 | 5.02 | 240.82 | 0.84 | 0.54 |
| SiT-XL/2-G [23] (cfg=1.50) | 2.62 | 4.18 | 252.21 | 0.84 | 0.57 |
| **FlowDCN-XL/2(cfg=1.375, ODE-50)** | 2.76 | 5.29 | 240.6 | 0.83 | 0.51 |
| **FlowDCN-XL/2(cfg=1.375, SDE-250)** | **2.44** | **4.53** | **252.8** | 0.84 | 0.54 |

Table 5: **Benchmarking class-conditional image generation on ImageNet 512×512.** Our FlowDCN-XL/2 is fine-tuned for 100k steps from the same model trained on $256 \times 256$ resolution setting of 1.5M steps

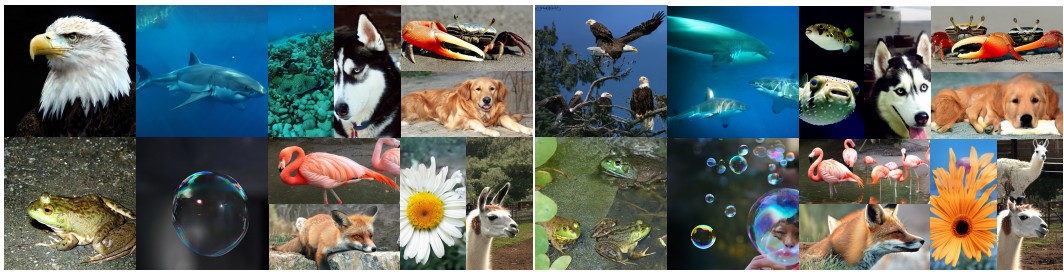

with $S_{\max}$ Adjustment                     without $S_{\max}$ Adjustment

Figure 4: **Visualization Comparison about $S_{\max}$ Adjustment.** Here are the $512 \times 512, 256 \times 512$ and $512 \times 256$, three type resolution images. We employ the same latent noise as start, sampling with Euler SDE solver for 250 steps. With $S_{\max}$ Adjustment, sampled images consistently looks better.

FlowDCN with 384M training images of Tab. 4). Although fine-tuned with limited 100k steps, our FlowDCN demonstrated powerful performance.

**Comparison with other generative models.** We report the final metrics of FlowDCN-XL/2 on 512 × 512 ImageNet Dataset at Tab. 5. Our FlowDCN achieves much better FID and sFID performance compared to its counterparts. Using Euler SDE solver with 250 steps and classifier-free guidance with 1.375, our FlowDCN obtains 4.53 sFID and 2.44 FID results. Using Euler ODE solver with 50 steps and classifier-free guidance with 1.375, our FlowDCN obtains 5.29 sFID and 2.76 FID result. As shown in Tab. 5, our FlowDCN achieves better sFID and captures better spatial structure distributions.

## 4.4   Arbitrary Resolution Extension

For the resolution extrapolation evaluation, we follow the setting in FiT. We select 320x320 and 224x448 as the evaluation arbitrary resolution. It is worth noting that our FlowDCN can handle arbitrary resolution within a reasonable range, the reasonable range is determined by the training setting and training dataset. As our primary goal is to explore DCN-like architectures in universal image generation, we do not intend to enhance the resolution extrapolation nature by data processing. Therefore, we do not employ any multiple aspect ratio training techniques like FiT[18]. Instead, we directly use the FlowDCN model trained on the center-cropped 256x256 ImageNet dataset for arbitrary resolution extension experiments, showcasing the model's inherent capabilities. Moreover, we provide resolution extension experiments with various aspect ratio training techniques in the Appendix.

**Metric comparsion.** We report the evaluation results on Tab. 6. For Base-size models, our FlowDCN-B/2 achieves much better results on 320x320 resolution, with 34.4 FID and 35.7 FID using $S_{\max}$ adjustment, outperforming FiT and DiT with a large margin. On 224x448 resolution, our FlowDCN-

| Method | 320×320 (1:1) | | | 224×448 (1:2) | | |
|---|---|---|---|---|---|---|
| | FID↓ | sFID↓ | IS↑ | FID↓ | sFID↓ | IS↑ |
| DiT-B | 95.5 | 108.7 | 18.4 | 109.1 | 110.7 | 14.0 |
| DiT-B$_{EI}$ | 81.5 | 62.3 | 21.0 | 133.2 | 72.5 | 11.1 |
| DiT-B$_{PI}$ | 72.5 | 54.0 | 24.2 | 133.4 | 70.3 | 11.7 |
| FiT-B | 61.4 | 30.7 | 31.0 | 44.7 | **24.1** | 37.1 |
| FiT-B$_{vYaRN}$ | 44.8 | 38.0 | 44.7 | **41.9** | 42.8 | **45.9** |
| FiT-B$_{vNTK}$ | 57.3 | 31.3 | 34.0 | 43.8 | 26.3 | 39.2 |
| **FlowDCN-B/2** | **34.4** | **27.2** | **52.2** | 71.7 | 62.0 | 23.7 |
| + $S_{max}$ **Adjust** | 35.7 | 29.3 | 51.2 | 81.1 | 40.2 | 21.1 |

(a) **Metrics Results on Base-Size Models**

| Method | 320×320 (1:1) | | | 224×448 (1:2) | | |
|---|---|---|---|---|---|---|
| | FID↓ | sFID↓ | IS↑ | FID↓ | sFID↓ | IS↑ |
| ADM-G,U [10] | 9.39 | **9.01** | 162 | 11.34 | **14.5** | 146 |
| LDM-4 [40] | 6.24 | 13.21 | 220 | 8.55 | 17.62 | 186 |
| UViT-H/2 [11] | 7.65 | 16.30 | 208 | 67.1 | 42.92 | 45.5 |
| MDT-G [41] | 383 | 136 | 4.24 | 365 | 142.8 | 4.91 |
| DiT-XL/2 [12] | 9.98 | 23.57 | 225 | 94.94 | 56.06 | 35.7 |
| FiT-XL/2 [18] | **5.42** | 15.41 | 252 | **7.9** | 19.63 | **215** |
| **FlowDCN-L/2** | 5.99 | 9.71 | 238 | 12.8 | 17.9 | 168 |
| **FlowDCN-XL/2** | 5.86 | 13.5 | **275** | 12.9 | 20.6 | 184 |

(b) **Metrics Results on Large-Size Models**

Table 6: **Benchmarking resolution extrapolations on ImageNet dataset.** On the Base-size Models benchmark, our FlowDCN achieves much better results on 320x320 resolution and comparable results on 224x448 resolution. On the Large-Size Models benchmark, our flowDCN shows comparable extrapolation performance to SoTA models.

B/2 achieves comparable results. Note our FlowDCN not employs any various aspect ratio training in Tab. 6, so we believe our FlowDCN-B/2 can achieve better results when incorporating such training augmentations. For large-size models, we report our FlowDCN-L/2 and FlowDCN-XL/2 with $S_{max}$ adjustment in Tab. 6b, our model shows comparable results to SoTA models. Meanwhile, we notice that FiT performs poorly on 256x256 resolution in Tab. 4, which we hypothesize is due to resolution-related data augmentation hurting the fitting power of original resolution distributions. Furthermore, as FiT employs the $256 \times 256$ reference statistics from ADM Eval Suite[10] to evaluate all resolution(even for $224 \times 448$), we suspect this evaluation paradigm is unreasonable.

**Visual quality comparison of $S_{max}$.** In Tab. 6a, We notice FlowDCN-B/2 with $S_{max}$ adjustment does not exhibit better results than directly generating images, we hypothesize that FID and sFID are low level visual quality assessments, not reflecting semantic visual quality. So we also provide the visualization comparisons of our FlowDCN-XL/2 with and without $S_{max}$ Adjustment in Fig. 4 and Appendix. With $S_{max}$ Adjustment, generated images consistently look better. But not all the cases demand $S_{max}$ Adjustment, some images like the bubble and the husky case in Fig. 4, still look good even without $S_{max}$ Adjustment. More comparison examples can be found in the Appendix.

## 5    Conclusion

In this paper, we have presented FlowDCN, a novel deformable convolutional network for arbitrary-resolution image generation. Our FlowDCN model leverages the strengths of both group-wise multiscale deformable convolutions and linear flow to generate high-quality images of various resolutions with high flexibility. Through extensive experiments, we demonstrate that FlowDCN outperforms the state-of-the-art transformer-based counterparts in terms of performance, convergence speed, and computational efficiency. Additionally, our model exhibits strong resolution extrapolation capabilities, achieving comparable results to previous models on arbitrary resolution without any additional training techniques. We believe that FlowDCN has a great potential to become a powerful tool for a wide range of image generation tasks and applications.

## Limitations and Future Works

Our current implementation of deformable convolution backward is inefficient to be on par with Attention. Our primary focus remains on optimizing the training speed. Once we have made significant strides in training optimization, we plan to scale up our FlowDCN to accommodate larger model parameters and higher training resolution, paving the way for more advanced explorations.

## Acknowledgement

This work is supported by the National Key R&D Program of China (No. 2022ZD0160900), the National Natural Science Foundation of China (No. 62076119), the Fundamental Research Funds for the Central Universities (No. 020214380119), the Nanjing University-China Mobile Communications Group Co., Ltd. Joint Institute, and the Collaborative Innovation Center of Novel Software Technology and Industrialization.

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

## A. Model Details

| Model | Layers $N$ | Hidden size $d$ | Groups |
|-------|-----------|-----------------|--------|
| FlowDCN-S | 12 | 384 | 6 |
| FlowDCN-B | 12 | 768 | 12 |
| FlowDCN-L | 24 | 1024 | 16 |
| FlowDCN-XL | 28 | 1152 | 16 |

Table 7: **Details of FlowDCN models.** We follow DiT for the Small (S), Base (B), Large (L) and XLarge (XL) model configurations.

## B. Comparisons between FlowCNN and FlowDCN on ImageNet $256 \times 256$

**The relationship between DCN and common CNN.** As Eq. (6) states, DCN introduces a deformable field $\Delta p(x)$ and dynamic weight $w(x)$. When all features shares the same static weight instead of dynamic, and deformable field $\Delta p(x)$ degrades to zeros, DCN degenerates to common CNN. Therefore, in most scenarios, DCN-like architectures are more powerful than common CNNs. Furthermore, the *fix $p_k$* in Tab. 2c indicates that we freeze the $p_k$ (not the deformable field $\Delta p(x)$) and initialize it with a predefined grid.

**Why not try a common CNN architecture.** In many computer vision tasks, traditional CNNs have been outperformed by transformers, so we opted to explore the modern, advanced CNN variant, Deformable Convolutional Networks (DCN). Additionally, we conducted a small experiment where we replaced the DCN block in FlowDCN with standard 3x3 and 5x5 group-wise convolution blocks.

| Model | layers | groups | channels | Params (M) | FID | sFID | IS |
|-------|--------|--------|----------|------------|-----|------|-----|
| SiT-S/2 | 12 | 6 | 384 | 33.0 | 57.64 | 9.05 | 24.78 |
| FlowCNN-3x3 | 12 | 8 | 512 | 49.1 | 59.0 | 10.7 | 27.4 |
| FlowCNN-5x5 | 12 | 6 | 384 | 33.1 | 63.0 | 10.9 | 23.6 |
| FlowDCN-S/2 | 12 | 6 | 384 | 30.3 | 54.6 | 8.8 | 26.4 |

Table 8: **Image generation metrics comparisons between SiT, FlowDCN and FlowCNN under 400k training steps budgets**.

| Method | 256×256 (1:1) | | | 320×320 (1:1) | | | 224×448 (1:2) | | | 160×480 (1:3) | | |
|--------|------|------|------|------|------|------|------|------|------|------|------|------|
| | FID↓ | sFID↓ | IS↑ | FID↓ | sFID↓ | IS↑ | FID↓ | sFID↓ | IS↑ | FID↓ | sFID↓ | IS↑ |
| DiT-B | 44.83 | 8.49 | 32.05 | 95.47 | 108.68 | 18.38 | 109.1 | 110.71 | 14.00 | 143.8 | 122.81 | 8.93 |
| DiT-B + EI | 44.83 | 8.49 | 32.05 | 81.48 | 62.25 | 20.97 | 133.2 | 72.53 | 11.11 | 160.4 | 93.91 | 7.30 |
| DiT-B + PI | 44.83 | 8.49 | 32.05 | 72.47 | 54.02 | 24.15 | 133.4 | 70.29 | 11.73 | 156.5 | 93.80 | 7.80 |
| FiT-B | 36.36 | 11.08 | 40.69 | 61.35 | 30.71 | 31.01 | 44.67 | 24.09 | 37.1 | 56.81 | 22.07 | 25.25 |
| FiT-B + VisionYaRN | 36.36 | 11.08 | 40.69 | 44.76 | 38.04 | 44.70 | 41.92 | 42.79 | 45.87 | 62.84 | 44.82 | 27.84 |
| FiT-B + VisionNTK | 36.36 | 11.08 | 40.69 | 57.31 | 31.31 | 33.97 | 43.84 | 26.25 | 39.22 | 56.76 | 24.18 | 26.40 |
| FlowDCN-B | 28.5 | 6.09 | 51 | 34.4 | 27.2 | 52.2 | 71.7 | 62.0 | 23.7 | 211 | 111 | 5.83 |
| FlowDCN-B (+VAR) | 23.6 | 7.72 | 62.8 | 29.1 | 15.8 | 69.5 | 31.4 | 17.0 | 62.4 | 44.7 | 17.8 | 35.8 |
| + $S_{max}$ Adjust | 23.6 | 7.72 | 62.8 | 30.7 | 19.4 | 68.5 | 37.8 | 22.8 | 54.4 | 53.3 | 22.6 | 31.5 |

Table 9: **Benchmarking resolution extrapolations on ImageNet with various aspect ratio training**. VAR indicates various aspect ratios training. We follow the same evaluation pipeline of FiT without using CFG.

## C. Resolution Extension with Various Aspect Ratios Training

While FlowDCN, trained on fixed-resolution images, is capable of generating images of arbitrary resolution within a reasonable aspect ratio range, its performance can be improved by adopting variable aspect ratio (VAR) training instead of a fixed 256x256 resolution. To ensure a fair comparison with FiT, which inherently uses VAR, we train a FlowDCN-B/2 model from scratch using VAR techniques. We evaluate our model using the same pipeline and reference batch as FiT, without CFG.

