# OpenReview forum: "Exploring DCN-like architecture for fast image generation with arbitrary resolution"
_NeurIPS.cc/2024/Conference — NeurIPS 2024 poster_

### Official Review · Reviewer_LWE8 · 2024-07-12

**Soundness:** 3
**Presentation:** 3
**Contribution:** 3
**Rating:** 5
**Confidence:** 5

**Summary:**

This paper presents a new convolution-based diffusion model that is able to generate images at arbitrary resolutions when trained on fixed resolutions. It also achieves comparable performance when trained and tested on the same resolution compared to transformer backbones such as U-VIT and DiT.

**Strengths:**

- This model is efficient: it achieves similar performance compared to SOTA methods but with 20% lower latency, 8% fewer parameters, and approximately 20% fewer floating-point operations.
- It enables arbitrary-resolution generation, which is an interesting achievement. Previously, we needed to fine-tune diffusion models on specific resolutions to get good images. Now this enables more flexible generation.

**Weaknesses:**

- As I mentioned, what I like most about this paper is the ability for arbitrary resolution generation. However, I think more experiments regarding handling arbitrary resolutions are needed. For example, it is not very clear which part enables the proposed method to handle arbitrary resolutions. In the contribution section, the authors attribute this ability to Scale Adjustment. Does the MultiScale DCN Block also help? However, in Line 245, the authors say this ability is determined by the training setting and training dataset, which is also confusing since the training data is 256x256 ImageNet.
- In Table 4, it would be good to see what happens if this model is trained for more iterations. Currently, the model doesn’t beat SiT-XL/2 on ImageNet 256. I wonder if the model will get better results when trained for longer iterations. (My personal experience on DCN is that it sometimes converges faster but not necessarily better.)
- One interesting benefit of the proposed model is the ability to generate images at any resolution. However, the results are worse than FiT on the 1:2 setting (Table 5). It would be great to see if the performance improves when multiple aspect ratio training augmentations are used. In addition, what reference statistics are you using for the 1:2 setting? And what is the performance when you also use 256×256 reference statistics?

**Questions:**

Please see the question in the weaknesses section. I will be willing to raise my rating if these questions are resolved.

**Limitations:**

Yes, discussed.

---

> ### Author Rebuttal · Authors · 2024-08-04
>
> **Which part contributes most to arbitrary resolution generation?**
> Sorry for the confusion. Basically, the fundamental contribution to the ability of arbitrary resolution generation is our MultiScale DCN Block, which equips our model with the flexibility to handle different resolutions. So, the sentence in Line 245 is miss-leading and will be revised in the final version. Smax adjustment is another technique to further improve the global semantic consistency of generated images, as observed in Fig. 4. In rebuttal experiment, we further find aspect ratio augmentation during training is helpful to improve its performance on arbitary resolution generation. This result is easy to understand as data augmentation typically contributes to a better model with higher performance.
>
> **ImageNet $256\times256$ Long Iterations Experiments.**
> Notably, our FlowDCN-XL/2 model, trained for only 1.5 million steps, achieves comparable results to SiT, which was trained for 7 million steps. This raises the question of whether FlowDCN-XL/2 is inherently more powerful or simply converges faster. To clarify this, we extended the training of our FlowDCN-XL/2 model by an additional 400k steps. The results show that our model attains a FID score of 2.01 with the ODE solver and 2.00 with the SDE solver, significantly outperforming SiT.
>
> | Generative Models           | Total Images（M） | Total GFLOPs                   | FID $\downarrow$ | sFID $\downarrow$ | IS $\uparrow$   | P $\uparrow$  | R $\uparrow$  |
> |-------------------------------------------------|-----------------|--------------------------------|------------------|-------------------|-----------------|---------------|---------------|
> | ADM-U                                           | 507             | $3.76\times10^{12}$            | 3.60             | -                 | 247.67          | 0.87          | 0.48          |
> | LDM-4                                           | 213             | $2.22\times10^{10}$            | 3.95             | -                 | 178.22          | 0.81          | 0.55          |
> | U-ViT-H/2                                       | 512             | $6.81\times10^{10} $           | 2.29             | -                 | 247.67          | {0.87}        | 0.48          |
> | DiT-XL/2                                        | 1792            | $2.13\times10^{11}$            | 2.27             | 4.60              | 278.24 | 0.83          | 0.57          |
> | DiffusionSSM-XL                                 | 660             | $1.85\times10^{11}$            | 2.28             | 4.49              | 259.13          | 0.86 | 0.56          |
> | SiT-XL/2                                        | 1792            | $2.13\times10^{11}$            | 2.06         | 4.50              | 270.27          | 0.82          | 0.59 |
> | FiT-XL/2                                        | 450             | -                              | 4.27             | 9.99              | 249.72          | 0.84          | 0.51          |
> | FlowDCN-XL/2  (cfg=1.375; ODE) | 384    | ${3.57\times10^{10}}$ | 2.13           | 4.30     | 243.46        | 0.81       | 0.57       |
> | FlowDCN-XL/2 (cfg=1.375; SDE) | 384   | ${3.57\times10^{10}}$ | 2.08          | 4.38            | 257.53        | 0.82      | 0.57        |
> |FlowDCN-XL/2 (cfg=1.375; ODE) | 486    | ${4.52\times10^{10}}$ | 2.01          | 4.33            | 254.36        | 0.81        | 0.58     |
> | FlowDCN-XL/2 (cfg=1.375; SDE) | 486    | ${4.52\times10^{10}}$ | 2.00    | 4.37            | 263.16        | 0.82        | 0.58      |
>
>
> **Various Aspect Ratios training Experiments.** While FlowDCN, trained on fixed-resolution images, is capable of generating images of arbitrary resolution within a reasonable aspect ratio range, its performance can be improved by adopting variable aspect ratio (VAR) training instead of a fixed 256x256 resolution. To ensure a fair comparison with FiT, which inherently uses VAR, we train a FlowDCN-B/2 model from scratch using VAR techniques. We evaluate our model using the same pipeline and reference batch as FiT, without CFG.
>
> |                       | 256x256 FID | sFID   | IS      | 320x320 FID | sFID    | IS      | 224x448 FID | sFID    | IS      | 160x480 FID | sFID    | IS      |
> |-----------------------|-------------|--------|---------|-------------|---------|---------|-------------|---------|---------|-------------|---------|---------|
> | DiT-B                 | 44.83       | 8.49 | 32.05   | 95.47       | 108.68  | 18.38   | 109.1       | 110.71  | 14.00   | 143.8       | 122.81  | 8.93    |
> | with EI            | 44.83       | 8.49 | 32.05   | 81.48       | 62.25   | 20.97   | 133.2       | 72.53   | 11.11   | 160.4       | 93.91   | 7.30    |
> | with PI            | 44.83       | 8.49 | 32.05   | 72.47       | 54.02   | 24.15   | 133.4       | 70.29   | 11.73   | 156.5       | 93.80   | 7.80    |
> | FiT-B (+VAR)          | 36.36     | 11.08  | 40.69 | 61.35       | 30.71 | 31.01   | 44.67       | 24.09 | 37.1    | 56.81       | 22.07 | 25.25   |
> | with VisionYaRN | 36.36     | 11.08  | 40.69 | 44.76     | 38.04   | 44.70 | 41.92     | 42.79   | 45.87 | 62.84       | 44.82   | 27.84 |
> | with VisionNTK  | 36.36     | 11.08  | 40.69 | 57.31       | 31.31   | 33.97   | 43.84       | 26.25   | 39.22   | 56.76     | 24.18   | 26.40   |
> | FlowDCN-B             | 28.5        | 6.09   | 51      | 34.4        | 27.2    | 52.2    | 71.7        | 62.0    | 23.7    | 211         | 111     | 5.83    |
> | FlowDCN-B (+VAR)      | 23.6        | 7.72   | 62.8    | 29.1        | 15.8    | 69.5    | 31.4        | 17.0    | 62.4    | 44.7        | 17.8    | 35.8    |
> | with $S_{max}$ adjust | 23.6        | 7.72   | 62.8    | 30.7        | 19.4    | 68.5    | 37.8        | 22.8    | 54.4    | 53.3        | 22.6    | 31.5    |
>
> **The Evaluation of Arbitrary Resolution** We follow the same evaluation pipeline of FiT,  using the same reference statistics of ImageNet 256x256, without CFG.

---

### Official Review · Reviewer_yBFS · 2024-07-12

**Soundness:** 3
**Presentation:** 1
**Contribution:** 2
**Rating:** 5
**Confidence:** 5

**Summary:**

This paper introduces FlowDCN, a novel image generation model that efficiently generates high-quality images at various resolutions. The model's core innovation is a group-wise multiscale deformable convolution block, which enhances its adaptability to various resolutions. Built on flow-based generative models, FlowDCN demonstrates improved flexibility compared to DiT, FiT, SiT. The group-wise multiscale deformable convolution operation proves more efficient than conventional attention operation and DCNv4 operation. Additionally, the model incorporates a straightforward yet effective scale adjustment method for resolution extrapolation.

**Strengths:**

I have summarized the following two strengths of this paper:
1. **Techinical Contributions**:
The originality of this work lies in the creative application of deformable convolution to the image generation domain. By rethinking the design of this core building block, the authors have developed a new generative architecture that outperforms existing approaches. This is a novel Techinical contribution, as prior work has primarily focused on adapting transformer-based models for generation tasks.

2. **Experiment Results are Complete**:
The authors provide a comprehensive evaluation of their new deformable convolution operator, thoroughly examining the impact of each component. They present results for FlowDCN across various resolution settings (256x256, 320x320, 224x448), demonstrating its versatility. Additionally, the paper offers an efficiency analysis comparing their deformable convolution operator with the traditional attention operator and the DCNv4 operator. The authors also provide a detailed analysis of FlowDCN's computational requirements, including FLOPs, latency, and parameter count, offering valuable insights into the model's performance and resource utilization.

**Weaknesses:**

I think there are two main weaknesses of this paper:

1. **Presentation**:
Firstly, I recommand the authors should consider restructuring the paper to include a separate preliminary section. This section should cover the background on Deformable Convolution Revisited (Sec 2.1) and Linear-based flow matching (Sec 2.3), as these are not novel contributions of this work but rather foundational concepts. Secondly, the addition of a related work section would greatly benefit the reader. This section should provide a concise overview of relevant literature on generative models and vision backbone architectures. Lastly, the paper would benefit from a thorough proofreading to address typographical errors. For instance, in Appendix E, the title incorrectly uses "Uncarted" instead of "Uncurated". A careful review of the entire manuscript would help eliminate such oversights and enhance the overall quality of the presentation.

2. **Experiment**: The authors conducted their ablation studies primarily on the CIFAR dataset. However, it would be beneficial to extend these studies to larger datasets, such as ImageNet-1K, following the DiT. This expansion is crucial because performance characteristics can vary significantly between small and large datasets, potentially offering more comprehensive insights into the model's behavior. Furthermore, to facilitate a direct comparison with the FiT, it is recommended that the authors include evaluations at additional resolutions, specifically 160x480, 160x320, and 128x384. These additional evaluations would provide a more complete and robust comparison and offer a broader perspective on the model's performance across various image sizes.

**Questions:**

My question is mainly about the presentation and experiment of this paper, which have been described in detail in Weakness.

**Limitations:**

Yes, the authors candidly acknowledge several limitations in their study. First, they note that the current implementation of the backward pass for deformable convolution is inefficient, indicating potential for future optimization. Second, the research lacks experiments with higher resolution images, particularly at 512x512 pixels, which could provide valuable insights into the model's performance on larger visual data. Finally, the authors recognize the absence of experiments with larger model variants, which limits the exploration of FlowDCN's scalability. These identified shortcomings not only demonstrate the authors' critical assessment of their work but also highlight promising directions for future research and improvements to the model.

---

> ### Author Rebuttal · Authors · 2024-08-04
>
> **Presentation issues.**
> Thanks for your suggestions. We will re-organize the structure of our paper in the final version. We will add a preliminary section to introduce the background on DCN and Flow Matching. We will also rewrite the related work section to comprehensively discuss the relevant literature. The final version will be carefully revised according to your comments.
>
> **Ablation issues.**
> Thanks for your suggestion. We will conduct ablation studies on ImageNet256 to offer more comprehensive results into the model's behavior. Due to limited time, we cannot provide the detailed results before the rebuttal ddl.
>
> **ImageNet $512\times512$ Experiments.**
> We try to provide the ImageNet512 results of FlowDCN-XL/2 to show the generation power on high-resolution images. Due to the time constraint on the rebuttal, training FlowDCN-XL/2 from scratch on ImageNet at $512\times512$ resolution is not feasible. Instead, we fine-tune our pre-trained FlowDCN-XL/2 model, which was trained on ImageNet at $256\times256$ resolution of 1.5M steps, for just 100k steps on ImageNet 512. We adopt the same training pipeline as the original $256\times256$ resolution setting, without incorporating advanced techniques such as lognorm sampling, aspect ratio augmentation, and random cropping. Notably, our approach achieves a remarkable 2.76 FID score with the ODE solver in 50 steps, and 2.44 FID with the SDE solver in 250 steps.
> | Model                                     | FID$\downarrow$ | sFID$\downarrow$ | IS$\uparrow$   | Precision$\uparrow$ | Recall$\uparrow$ |
> |-------------------------------------------|-----------------|------------------|----------------|---------------------|------------------|
> | BigGAN-deep                               | 8.43            | 8.13             | 177.90         | 0.88                | 0.29             |
> | StyleGAN-XL                               | 2.41            | 4.06             | 267.75         | 0.77                | 0.52             |
> | ADM                                       | 23.24           | 10.19            | 58.06          | 0.73                | 0.60             |
> | ADM-U                                     | 9.96            | 5.62             | 121.78         | 0.75                | 0.64           |
> | ADM-G                                     | 7.72            | 6.57             | 172.71         | 0.87              | 0.42             |
> | ADM-G, ADM-U                              | 3.85            | 5.86             | 221.72         | 0.84                | 0.53             |
> | DiT-XL/2                                | 12.03           | 7.12             | 105.25         | 0.75                | 0.64           |
> | DiT-XL/2-G (cfg=1.50)                   | 3.04          | 5.02           | 240.82       | 0.84                | 0.54             |
> | **FlowDCN-XL/2(cfg=1.375, ODE-50)**  | 2.76            | 5.29             | 240.6          | 0.83                | 0.51             |
> | **FlowDCN-XL/2(cfg=1.375, SDE-250)** | **2.44**   | **4.53**    | **252.8** | 0.84                | 0.54             |
>
> **Various Aspect Ratios training Experiments.** Although FlowDCN, trained on fixed-resolution images, can generate images of arbitrary resolution within a reasonable aspect ratio range, the quality can be further enhanced by employing variable aspect ratio (VAR) training instead of a fixed $256\times256$ resolution. Therefore, we train a FlowDCN-B/2 model from scratch using VAR techniques to ensure a fair comparison with FiT, which inherently adopts VAR. We follow the same evaluation setting of FiT,  using the same reference batch, without CFG. Note that the generated resolutions are the same with the FiT.
>
> |                       | 256x256 FID | sFID   | IS      | 320x320 FID | sFID    | IS      | 224x448 FID | sFID    | IS      | 160x480 FID | sFID    | IS      |
> |-----------------------|-------------|--------|---------|-------------|---------|---------|-------------|---------|---------|-------------|---------|---------|
> | DiT-B                 | 44.83       | 8.49 | 32.05   | 95.47       | 108.68  | 18.38   | 109.1       | 110.71  | 14.00   | 143.8       | 122.81  | 8.93    |
> | with EI            | 44.83       | 8.49 | 32.05   | 81.48       | 62.25   | 20.97   | 133.2       | 72.53   | 11.11   | 160.4       | 93.91   | 7.30    |
> | with PI            | 44.83       | 8.49 | 32.05   | 72.47       | 54.02   | 24.15   | 133.4       | 70.29   | 11.73   | 156.5       | 93.80   | 7.80    |
> | FiT-B (+VAR)          | 36.36     | 11.08  | 40.69 | 61.35       | 30.71 | 31.01   | 44.67       | 24.09 | 37.1    | 56.81       | 22.07 | 25.25   |
> | with VisionYaRN  | 36.36     | 11.08  | 40.69 | 44.76     | 38.04   | 44.70 | 41.92     | 42.79   | 45.87 | 62.84       | 44.82   | 27.84 |
> | with VisionNTK   | 36.36     | 11.08  | 40.69 | 57.31       | 31.31   | 33.97   | 43.84       | 26.25   | 39.22   | 56.76     | 24.18   | 26.40   |
> | FlowDCN-B             | 28.5        | 6.09   | 51      | 34.4        | 27.2    | 52.2    | 71.7        | 62.0    | 23.7    | 211         | 111     | 5.83    |
> | FlowDCN-B (+VAR)      | 23.6        | 7.72   | 62.8    | 29.1        | 15.8    | 69.5    | 31.4        | 17.0    | 62.4    | 44.7        | 17.8    | 35.8    |
> | with $S_{max}$ adjust | 23.6        | 7.72   | 62.8    | 30.7        | 19.4    | 68.5    | 37.8        | 22.8    | 54.4    | 53.3        | 22.6    | 31.5    |

---

### Official Review · Reviewer_zBRq · 2024-07-12

**Soundness:** 3
**Presentation:** 3
**Contribution:** 3
**Rating:** 8
**Confidence:** 4

**Summary:**

The paper proposes a novel convolutional-based generative model called FlowDCN. This model addresses the challenge of generation speed and arbitrary-resolution image generation, which remains difficult for transformer-based diffusion methods due to their quadratic computation cost and limited resolution extrapolation capabilities. FlowDCN introduces a group-wise multiscale deformable convolution block, enabling the model to handle varying resolutions efficiently. The paper claims that FlowDCN achieves state-of-the-art performance on the 256x256 ImageNet benchmark, surpassing transformer-based counterparts in terms of convergence speed, visual quality, parameter efficiency, and computational cost.

**Strengths:**

- This paper presents a novel approach by leveraging multiscale deformable convolution blocks, which is innovative compared to traditional transformer-based models.
- It demonstrates impressive performance on the 256x256 ImageNet benchmark, achieving state-of-the-art FID scores with reduced parameters and FLOPs compared to other models. The efficiency is also improved with linear time and memory complexity, making it suitable for fast image generation.

**Weaknesses:**

It would be great to test FlowDCN on higher resolution such as 512x512 or higher, to showcase the efficiency of using a convolution-based model.

**Questions:**

The FID in Table 5 seems too high, is that because the models are trained with 400k iter and without cfg?

**Limitations:**

The authors have discussed the limitations and that makes sense to me.

---

> ### Author Rebuttal · Authors · 2024-08-04
>
> **FID in Table 5 seems too high.** Yes, we follow SiT, FiT and DiT to train our FlowDCN under 400K budgets. We follow the evaluation pipeline of FiT to obtain the metrics of arbitrary resolution generation without using CFG.
>
>
> **ImageNet $512\times512$ Experiments.**
> Given the time constraint on the rebuttal, it is infeasible to train FlowDCN-XL/2 from scratch on ImageNet at a resolution of 512x512. As an alternative, we fine-tune our pre-trained FlowDCN-XL/2 model, which was initially trained on ImageNet at 256x256 resolution for 1.5 million steps, for an additional 100,000 steps on ImageNet 512. We utilize the same training pipeline as the original 256x256 resolution setting. Notably, our approach achieves a remarkable FID score of 2.76 with the ODE solver in just 50 steps, and 2.44 with the SDE solver in 250 steps.
>
> | Model                                     | FID$\downarrow$ | sFID$\downarrow$ | IS$\uparrow$   | Precision$\uparrow$ | Recall$\uparrow$ |
> |-------------------------------------------|-----------------|------------------|----------------|---------------------|------------------|
> | BigGAN-deep                               | 8.43            | 8.13             | 177.90         | 0.88                | 0.29             |
> | StyleGAN-XL                               | 2.41            | 4.06             | 267.75         | 0.77                | 0.52             |
> | ADM                                       | 23.24           | 10.19            | 58.06          | 0.73                | 0.60             |
> | ADM-U                                     | 9.96            | 5.62             | 121.78         | 0.75                | 0.64           |
> | ADM-G                                     | 7.72            | 6.57             | 172.71         | 0.87              | 0.42             |
> | ADM-G, ADM-U                              | 3.85            | 5.86             | 221.72         | 0.84                | 0.53             |
> | DiT-XL/2                                | 12.03           | 7.12             | 105.25         | 0.75                | 0.64           |
> | DiT-XL/2-G (cfg=1.50)                   | 3.04          | 5.02           | 240.82       | 0.84                | 0.54             |
> | **FlowDCN-XL/2(cfg=1.375, ODE-50)**  | 2.76            | 5.29             | 240.6          | 0.83                | 0.51             |
> | **FlowDCN-XL/2(cfg=1.375, SDE-250)** | **2.44**   | **4.53**    | **252.8** | 0.84                | 0.54             |

---

> > ### Comment · Reviewer_zBRq · 2024-08-13
> >
> > Thanks for the response. It would be great to showcase the latency for 512x512 as I expect the gap between DiT and FlowDCN would be larger since the Transformer-based model has a quadratic complexity for the resolution. Overall I still believe this paper is good and vote for accept.

---

> > > ### Author Response · Authors · 2024-08-14
> > > **Inference Latency**
> > >
> > > We appreciate your thoughtful responses and encouraging feedback. To provide a comprehensive comparison, we break down the total inference latency into its constituent parts, namely the MLP and DCN/Attn components, and examine each part's inference latency in detail. Specifically, we measure the inference time (in seconds) on an A10 GPU with a batch size of 16 and float32.
> > >
> > > Resolution-256x256:
> > > |  Model  | MLP  | DCN/Attn| Total|
> > > |  ----  | ----  |  ----  | ----  |
> > > | DiT-XL/2 | 0.2 |0.17|0.37|
> > > | FlowDCN-XL/2 | 0.2 |0.10(-41%)|0.30|
> > >
> > > Resolution-512x512:
> > > |  Model  | MLP  | DCN/Attn| Total|
> > > |  ----  | ----  |  ----  | ----  |
> > > | DiT-XL/2 | 0.93 |1.07|2.0|
> > > | FlowDCN-XL/2 | 0.93 |0.48(-55%)|1.41|

---

### Official Review · Reviewer_rZWQ · 2024-07-13

**Soundness:** 3
**Presentation:** 3
**Contribution:** 3
**Rating:** 6
**Confidence:** 4

**Summary:**

This paper presents a purely deformable convolution based architecture for flow-matching based diffusion models. Such a purely convolution-based models can easily generalize to different aspect ratio/resolution during testing, which is a pain point for transformer-based models. Evaluated on ImageNet 256x256 benchmark, the proposed FlowDCN achieves promising results with fewer parameters/FLOPs and faster sampling speed.

**Strengths:**

- The paper is well written and easy to follow


- I like the simple idea of adopting convolution-based architecture to achieve a better generalization across different resolution, which seems a simple yet effective solution IMO.

- The adaptation of deformable convolution and corresponding optimization is technically solid.

- Extensive experiments and good performance

**Weaknesses:**

Although I like the idea of using a CNN style architecture for better generalization across different resolution, I do have several questions after reading the paper:

- In Tab. 2(c), if I understand correctly, the deformable conv degrades back to a normal conv when p_k and s(x) are both fixed. From the table it achieves a similar performance to the final model, which makes it less convincing to use deformable convolution, which requires additional CUDA optimization etc. Taking Tab. 2 (b) into consideration as well, it seems that a normal convolution with larger kernel can do as well. Please illustrate why deformable conv is especially needed. Although normal convolution is a local op, under the setting of input size 32 x 32 (Cifar, or ImaegNet with VAE latents), a normal conv with larger kernel seems "global" enough IMHO.

- Tab.3 and Tab.4 reports different FLOPs, which is confusing. I see Tab.4's GFLOPs seems refer to the total training compute. It would be better if they can be explicitly annotated to avoid confusion.

- Although FlowDCN has a comparable or better FID compared to SiT or DiT, its Inception score falls behind, can the author provide some in-depth analysis on this?

- One major advantage of conv-based models against transformer-based models is the low complexity when scaling up to larger resolution input. It would be great if the effectiveness of FlowDCN can be verified on ImageNet 512 x 512 as well besides 256 x 256. I understand the 512 experiments can be costly and do not expect to see them in the rebuttal, but would highly appreciate if at least the results of testing 256 x 256 models on 512 x 512 benchmarks can be reported, as "arbitrary resolution extension" is claimed to be one major advantages of FLowDCN.

**Questions:**

Please see weakness for detailed questions.

After reading the paper, although I have some concerns in the representation and experiments, I do appreciate the efforts on purely convolution architecture for diffusion models, which naturally generalizes better to different resolutions compared to transformer-based ones. Thus my initial rating is weak accept. I expect the author can justify the usage of deformable convolution based on results from Tab. 3 (c), and how would the model perform on ImageNet 512 benchmark under "resolution extension" setting.

**Limitations:**

No other limitations as far as I am aware.

---

> ### Author Rebuttal · Authors · 2024-08-04
>
> **The relationship between DCN and common CNN.**
> As Eq.3 states, DCN introduces a deformable field $\Delta p(x)$ and a dynamic weight $w(x)$. When all channels share the same static weights instead of dynamic ones, and deformable field $\Delta p(x)$ degrades to zeros, DCN degenerates to common CNN. Therefore, in general, DCN-like architectures are more flexible and powerful than common CNNs. It should be noted that, the *fixed $p_k$* in Tab. 2 (c) indicates that we freeze the $p_k$ (not the deformable field $\Delta p(x)$) and initialize it with a predefined grid. This setting is not equivalent to the common CNN.
>
>
> **Why not try a normal convolution.** In many computer vision tasks, traditional CNNs have been outperformed by transformers, so we opted to explore a modern and advanced variant of CNN: Deformable Convolutional Networks (DCN). Additionally, we conducted a small experiment on ImageNet 256, where we replaced the DCN block in FlowDCN with a standard 3x3 group-wise convolution block. Due to limited time, the experiments with larger kernel of 5*5 is still on the way.
> | Model         | Params (M) | FID$\downarrow$ | sFID$\downarrow$ | IS$\uparrow$ |
> |---------------|------------|-----------------|------------------|--------------|
> | SiT-S/2       | 33.0       | 57.64           | 9.05             | 24.78        |
> | FlowCNN-S/2   | 49.1       | 59.0            | 10.7             | 27.4         |
> | FlowDCN-S/2 | 30.3       | 54.6          | 8.8              | 26.4       |
>
> **Different annotations for Total Training GFLOPs and 1-NFE FLOPs in Tab3. & 4.** We appreciate your insightful feedback and concur with your opinions. We will clarify this point finally.
>
> **Lower IS metric.** In Tab. 3, our FlowDCN consistently achieves superior performance in terms of the IS metric compared to its counterparts. However, the final results reported in Tab. 4 surprisingly yield relatively lower IS scores compared to DiT and SiT. Notably, we observe a trend of IS improvement when extending the training iteration by an additional 400k steps, with the score increasing from 257.5 at 1.5M training steps to 263.26 at 1.9M training steps. This suggests that the limited number of training steps may be the primary cause of the lower IS metric.
>
> **ImageNet $512\times512$ Experiments.**
> Due to the time constraint on the rebuttal, training FlowDCN-XL/2 from scratch on ImageNet at $512\times512$ resolution is not feasible. Instead, we fine-tune our pre-trained FlowDCN-XL/2 model, which was trained on ImageNet at $256\times256$ resolution for 1.5M steps, for just 100k steps on ImageNet 512. We adopt the same training pipeline as the original $256\times256$ resolution setting, without incorporating advanced techniques such as lognorm sampling, aspect ratio augmentation, and random cropping. Notably, our approach achieves a remarkable 2.76 FID score with the ODE solver in 50 steps, and 2.44 FID with the SDE solver in 250 steps.
>
> | Model                                     | FID$\downarrow$ | sFID$\downarrow$ | IS$\uparrow$   | Precision$\uparrow$ | Recall$\uparrow$ |
> |-------------------------------------------|-----------------|------------------|----------------|---------------------|------------------|
> | BigGAN-deep                               | 8.43            | 8.13             | 177.90         | 0.88                | 0.29             |
> | StyleGAN-XL                               | 2.41            | 4.06             | 267.75         | 0.77                | 0.52             |
> | ADM                                       | 23.24           | 10.19            | 58.06          | 0.73                | 0.60             |
> | ADM-U                                     | 9.96            | 5.62             | 121.78         | 0.75                | 0.64           |
> | ADM-G                                     | 7.72            | 6.57             | 172.71         | 0.87              | 0.42             |
> | ADM-G, ADM-U                              | 3.85            | 5.86             | 221.72         | 0.84                | 0.53             |
> | DiT-XL/2                                | 12.03           | 7.12             | 105.25         | 0.75                | 0.64           |
> | DiT-XL/2-G (cfg=1.50)                   | 3.04          | 5.02           | 240.82       | 0.84                | 0.54             |
> | **FlowDCN-XL/2(cfg=1.375, ODE-50)**  | 2.76            | 5.29             | 240.6          | 0.83                | 0.51             |
> | **FlowDCN-XL/2(cfg=1.375, SDE-250)** | **2.44**   | **4.53**    | **252.8** | 0.84                | 0.54             |

---

> ### Author Response · Authors · 2024-08-12
> **A rectification of ‘Why not try a normal convolution’ section.**
>
> Deeply sorry, that the FlowCNN experiments referenced earlier utilized models with substantially larger parameter counts, attributable to a misconfigured number of groups and channels.
>
> We have conducted a small experiment on ImageNet 256, where we replaced the DCN block in FlowDCN with a standard 3x3/5x5 group-wise convolution block, donated as FlowCNN. The results are as follows.
>
> |   |  |  |  |   |   |   |   |
> |---|---|---|---|---|---|---|---|
> |Model | layers | groups | channels |Params (M)| FID | sFID | IS
> |SiT-S/2 | 12| 6| 384 |33.0| 57.64|9.05|24.78|
> |FlowCNN-3x3| 12| 8| 512 |49.1|59.0|10.7|27.4|
> |FlowCNN-5x5| 12| 6| 384 |33.1|63.0|10.9|23.6|
> |FlowDCN-S/2| 12| 6| 384 | 30.3|54.6|8.8|26.4|

---

### Decision · Program_Chairs · 2024-09-25

**Decision:**

Accept (poster)

**Comment:**

This work presents FlowDCN, a generative model using pure convolutions, enabling arbitrary resolution image generation. Additionally, the authors propose a new design of learnable group-wise deformable convolution block. Promising results have been demonstrated on the ImageNet generation benchmark.

Initially, the reviewers were concerned about the relationship between DCN and common CNN, experiments on ImageNet 512x512 (FID and latency), paper presentation, and so on. During the reviewer-author discussion period, the provided rebuttal successfully assuaged the reviewers' concerns. In the end, the paper received all accept recommendations (2 borderline accepts, 1 weak accept and 1 strong accept). After considering the author rebuttal and reviewer discussion/comments, the area chair agrees with this recommendation.

Finally, the authors are strongly encouraged to incorporate the rebuttal and reviewer suggestions to their final camera-ready version.